# Development of a Recycling Process and Characterization of EVA, PVDF, and PET Polymers from End-of-Life PV Modules

**DOI:** 10.3390/ma17040821

**Published:** 2024-02-08

**Authors:** Marek Królikowski, Michał Fotek, Piotr Żach, Marcin Michałowski

**Affiliations:** 1Faculty of Chemistry, Warsaw University of Technology, Noakowskiego 3, 00-664 Warsaw, Poland; 2Faculty of Automotive and Construction Machinery, Warsaw University of Technology, Narbutta 84, 02-524 Warsaw, Polandpiotr.zach@pw.edu.pl (P.Ż.); 3Faculty of Mechatronics, Warsaw University of Technology, św. Andrzeja Boboli 8, 02-525 Warsaw, Poland; marcin.michalowski@pw.edu.pl

**Keywords:** photovoltaic module, recycling PV module, chemical treatment, mechanical–thermal treatment, FTIR spectroscopy, elemental analysis, EVA, PET, PVDF

## Abstract

Photovoltaic (PV) modules are highly efficient power generators associated with solar energy. The rapid growth of the PV industry will lead to a sharp increase in the waste generated from PV panels. However, electro-waste can be successfully used as a source of secondary materials. In this study, a unique procedure for recycling PV modules was developed. In the first stage, the aluminum frame and junction box, 18wt%. and 1wt%. of the module, respectively, were removed. The following stage was crucial, involving a mechanical–thermal method to remove the glass, which accounts for 70wt%. As a result, only 11wt%. of the initial mass of the PV was subjected to the next stage of chemical delamination, which reduced the amount of solvent used. Toluene was used to swell the ethylene vinyl acetate, EVA, and allow for the separation of the PV module. The effects of temperature and ultrasound on separation time were investigated. After the separation of silicon cells, metal ribbons, EVA, and the backsheet were obtained. The purity of the polymers was determined by FTIR and elemental analysis. Thermal properties were measured using DSC calorimetry to determine the basic parameters of the material.

## 1. Introduction

The widespread use of traditional fossil fuels, as well as the resultant increased environmental pollution, has prompted a gradual transition in energy supply to renewable energy. Additionally, developing sustainable renewable energy is an effective strategy for reducing carbon emissions. Due to its non-toxic emissions and simple installation, photovoltaic power generation is gradually gaining dominance in renewable energy. The total global cumulative PV installed capacity was 1185 GW at the end of 2022 [1]. Compared to 2021, this is an increase of 240 GW of new systems installed and commissioned. The Chinese market continued to dominate new and cumulative capacity and added 106 GW. Europe demonstrated continued strong growth with 39 GW installed, led by Spain, with 8.1 GW, Germany, with 7.5 GW, Poland, with 4.9 GW, and the Netherlands, with 3.9 GW. The rapid growth of the photovoltaic industry will lead to a sharp increase in the waste that is generated from PV panels. The most common silicon solar cells have a 20–30 year lifespan on average. The amount of PV e-waste will rise to 60–78 million tons globally. Degraded PV modules are predicted to produce 10% of all electronic waste by the year 2050 [2]. However, electro-waste can be successfully used as a source of secondary materials. Recently, several research publications, particularly review papers on the recycling of PV modules, have been published [3,4,5,6]. In general, PV recycling involves three stages: (1) manual/mechanical disassembly; (2) delamination: thermal, physical/mechanical, chemical, or combined; and (3) separation into silicon, metal and polymer fractions [7]. To separate specific layers of PV modules, physical procedures, chemical swelling or dissolving processes, and pyrolysis can be applied [8]. The physical and mechanical approaches focus on crushing and sifting the materials of PV modules. The method is straightforward, but it does not achieve sufficient separation of distinct layers, lowering the resource recovery ratio. Because physical and mechanical approaches do not affect EVA characteristics, EVA bonding remains effective. As a result, eliminating EVA bonding to adequately separate the different layers is a precondition for adequately recovering the available resources in PV modules. According to relevant studies [9,10], pyrolysis is the most effective approach for directly eliminating EVA and separating different layers. However, fluorine compounds from the polyvinyl fluoride, PVF, or polyvinylidene fluoride, PVDF, of the backsheet can be released into the environment in a direct thermal process [11,12]. Furthermore, glass and the backsheet inhibit the release of low-molecular-weight organic compounds during EVA pyrolysis, which can result in the contamination of glass and silicon cells with decomposition hydrocarbons and further mixing of glass and solar cells, lowering resource recovery rates. The chemical swelling or dissolving methods appear to be a promising way of separating the different layers of a PV module due to their low energy consumption and high rate of separation [13,14,15]. In the first step, the tempered glass was recovered using an organic solvent. Additionally, EVA copolymer contaminants were removed from the PV cells by thermal decomposition. On the other hand, silicon was obtained through a chemical etching process using HF and HNO_3_ with a surfactant by removing metal impurities from the surface of the recovered PV cell. It was possible to generate a high silicon yield [16]. The backsheet can be removed mechanically or using ethanol or toluene vapor [12,15]. However, the chemical process has disadvantages. EVA typically swells excessively, causing solar cells to be crushed [17]. Organic solvents such as benzene, toluene, o-dichlorobenzene, trichloroethylene, tetrahydrofuran, methyl ethyl ketone, methyl isobutyl ketone, or petroleum benzine are used in the swelling process, but some of them are toxic or mutagenic and contain halogens. The processes require a large amount of solvent and a high temperature, *T* > 70–80 °C, close to the boiling point of the solvent at atmospheric pressure, and last a long time, over 1 h [16,17,18]. As a consequence, recycling PV modules can be costly and time-consuming.

This study presents an alternative methodology for the separation of PV modules after their end of life. At first, the aluminum frame, junction box, and connecting wires were mechanically separated. The initial phase, while not novel, stands out primarily due to simplicity, affordability, and expediency. Subsequently, within a thermomechanical procedure, a layer of broken tempered glass was separated, which is a crucial stage since glass limits contact of the organic solvent with the next copolymer EVA layer. Moreover, it is worth noting that glass constitutes up to 70wt%. of the PV module, as well as a significant volume. It is pointless to use excess solvent, which does not affect the glass. The next procedure involves a chemical method with toluene as a solvent applied to the delamination of PV cells and the backsheet. The process was carried out at *T* = 35 °C and in a time below *t* = 40 min to obtain a swollen EVA copolymer separated from the silicon cells, busbars, and backsheet. A significantly shorter delamination time for PV modules was obtained in comparison to the procedures described in the literature [19,20,21]. The organic solvent also caused the backsheet to separate into two distinct layers composed of PVDF and PET. In a further step, the obtained polymers were confirmed and characterized by FTIR spectroscopy, elemental analysis, EA, and differential scanning calorimetry, DSC. This work presents the development of a straightforward approach for the environmentally sustainable recovery of EVA, PET, and PVDF from wasted crystalline silicon PV modules. The recovered materials can be effectively reused in PV modules, as well as find applications in the packaging and textile sectors.

## 2. Materials and Methods

### 2.1. Photovoltaic Modules

In order to analyze the various layers of PV modules, PV sheet sections were prepared from standard 275 W commercial end-of-life PV modules (polycrystalline–silicon type) obtained from the commercial PV facility of DAH Solar Co., Ltd., Hefei, China. The first step in recycling PV modules is to remove the aluminum frame and junction box, 18%wt. and 1%wt., respectively. This is usually conducted mechanically, and similar methods are presented in the literature [22,23,24]. To obtain the lamination structure required for the study, metal pliers and screwdrivers were used in advance to remove the aluminum frame and junction box of the PV module.

In the next step, the glass layer was removed. It should be noted that in most recycled PV modules, the glass sheet is broken into pieces a few to several millimeters in diameter and the glass is not recovered as a whole sheet due to physical damage during the lifetime, the method of dismantling and transporting the PV modules, and damage during the removal of the frame and junction box. The PV module on the glass side was thermally treated using Steinel^®^ HL1920 hot air guns (Steinel, Bloomington, MN, USA) with 2000 W of power, an infinitely variable temperature adjustment from 80 to 600 °C, and airflow control. The PV sample was heated to 170 ± 5 °C, at which point the EVA became soft and glass could be recovered by applying mechanical force. The temperature of the PV module was monitored using a pyrometer Benetech GM700 (Shenzhen Jumaoyuan Science And Technology Co., Ltd., Shenzhen, China) with an accuracy ±1.5 °C. After removing the glass, the laminated sample was cut into 5 cm × 5 cm pieces using scissors. Figure 1 shows photos of the PV module, the resulting PV sheet sections obtained from it that were used during the experiment, and the removed glass.

### 2.2. Swelling of EVA and Separation Process

Prepared in the previous stage, laminated samples were next subjected to solvent treatment to swell and separate EVA, extract the PV cells and connecting wires (busbars and fingers), and separate the backsheet. According to our experimental findings and the available literature [13,16,17], toluene (CAS No. 108-88-3, supplier: Chempur, purity > 99%, Piekary Śląskie, Poland) was used to swell the polymer. The swelling processes were prepared in a jacketed and thermostatted glass vessel with a volume of 150 mL. The jackets were connected to the thermostatic water bath, Julabo CORIO CD-BC6 (JULABO GmbH, Seelbach, Germany), to maintain a constant temperature with an accuracy of *T* = 0.05 °C. The heterogeneous mixtures of laminated samples and solvent were vigorously stirred using a mechanical stirrer IKA Microstar 15 Control (IKA, Staufen, Germany) with an R 1355 centrifugal stirrer. The rotation speed of the stirrer was constant and equal to 500 rpm, which was enough to provide perfect contact between the laminated samples and the solvent. In the next part of the experiment, an ultrasonic bath, PROCLEAN 2.0M ECO (Ulsonicx, Berlin, Germany), with a capacity of 2 dm^3^ and ultrasound power of 60 W was used. The bath was filled halfway with distilled water. The delamination process was carried out in a 250 mL glass beaker placed in the bath. During the measurements, the temperature of the solution in the beaker was measured using a P 750 thermometer (Dostmann electronic GmbH, Wertheim, Germany) equipped with a PT100 sensor.

To determine the degree of delamination of PV modules during the EVA copolymer swelling process, 2 cm^3^ of solvent was sampled and subjected to analysis of physicochemical properties, including density and dynamic viscosity. The swelling process of the EVA copolymer entails partial dissolution, especially of its shorter, uncrosslinked chains. In addition, some small-molecule compounds derived from the EVA copolymer migrate into the solvent. Consequently, the density and dynamic viscosity of the solvent change during the process of PV module delamination. The dependence of the physicochemical properties of toluene as a function of the degree of PV module delamination is presented in Appendix A.

#### 2.2.1. Density

The density of the toluene solution was determined under ambient pressure using a vibrating tube densimeter—DMA 4500 M, Anton Paar, Graz, Austria. The densimeter is equipped with an automated adjustment mechanism to account for the viscosity of the liquid sample. Calibration was conducted using water that had undergone double distillation and degassing, as well as air that had been dried. The utilization of two integrated Pt 100 platinum thermometers enables the achievement of precise temperature control with an accuracy of 0.05 °C. The measurement uncertainty was estimated to be better than 5 × 10^−4^ g∙cm^−3^.

#### 2.2.2. Dynamic Viscosity

The dynamic viscosity was measured using a cone/plate rheometer, model DVNext-LV, AMETEK Brookfield (Middleborough, MA, USA), with a relative uncertainty of *u*_r_(*η*) = 0.03 mPa·s. The viscosity error was determined using the standards APN26E and APN75 from Paragon Scientific Ltd. (Prenton, UK), as well as n-tetradecane (Alfa Aesar, Graz, Austria, purity > 99%). All measurements were conducted at an atmospheric pressure of *p* = 100 kPa, with an uncertainty in the temperature measurement of 0.1 K. The viscosity measurement was conducted at a constant temperature, *T* = 25.0 °C.

### 2.3. Analysis Methods: FT-IR Spectrometry and Elemental Analysis

The Fourier transform infrared spectrum analyses of the polymers were carried out using a Nicolet iS5, Thermo Scientific (Waltham, MA, USA) Mid Infrared FT-IR spectrometer equipped with an iD7 ATR Optical Base. Immediately before the measurement, the ATR crystal with ethanol was cleaned and dried. Then, the samples of polymers were directly placed on the surface of the ATR. The wavenumber ranged from 3900 to 400 cm^−1^.

Elemental analysis (EA) was performed on a CHNS Analyzer VARIO EL III (Elementar Analysensysteme GmbH, Langenselbold, Germany).

### 2.4. Differential Scanning Calorimetry

The differential scanning calorimetry (DSC) technique was used to determine the temperature (*T*_g_) and heat capacity change in the glass transition (Δ*C_p_*_(*g*)_). The studies were carried out using a Mettler Toledo DSC 1 STARe System calorimeter fitted (Mettler Toledo, Toronto, ON, Canada) with a liquid nitrogen cooling system and set to heat-flux mode. The sample cell was constantly fluxed with high-purity nitrogen at a constant flow rate of 20 mL·min^−1^. The apparatus was calibrated with the 99.9999 mol% purity indium sample and with high-purity ethylbenzene, *n*-octane, *n*-decane, *n*-octadecane, *n*-eicosane, cyclohexane, biphenyl, and water. The calibration experiment was carried out with a 5 °C·min^−1^ heating rate in the temperature range from −95 °C to 200 °C.

## 3. Results and Discussion

The external components of the crystalline PV modules consist of the aluminum frame and junction box, which account for approximately 18%wt. of the total PV mass, and the glass, which comprises around 70%wt. The process of removing the frame and junction box can be considered rather straightforward from a mechanical standpoint. Furthermore, a considerable fraction of PV modules that are stored lack these components, primarily because the aluminum and copper materials present in the frame, cables, and junction box hold significant value and are removed before the PV module becomes a waste of electrical and electronic equipment, WEEE. The subsequent stage in the separation process of the photovoltaic (PV) modules, which involves the elimination of the tempered glass layer, holds significant importance. On the one hand, the heaviest fraction, the glass of the PV module, is removed, and on the other hand, the direct exposure and chemical treatment of the EVA lamination layer are enabled.

### 3.1. Thermal Separation of the Glass Layer

The process of delamination of PV modules at low temperatures, namely, those below the thermal breakdown point of EVA, is a complex and challenging phenomenon. The strong adhesion of the EVA copolymer to glass surfaces is due to the siloxane chemical bonds formed [25]. Silane-based coupling agents are commonly used across several industries as adhesion promoters, facilitating the bonding of organic polymers to inorganic (glass and Si) and organic (PVDF) substrates. These agents have gained recognition as a regular ingredient in encapsulating materials specifically designed for PV module applications [26]. The methoxysilyl groups present in silane undergo hydrolysis, resulting in the formation of silanols. These silanols then undergo condensation on the hydroxylated glass surface, leading to the formation of siloxane bonds. The silane compound also exhibits affinity toward EVA and undergoes polymerization with the polymer itself, resulting in the formation of an interphase region, as depicted in Figure 2.

The samples of PV modules on the glass side were thermally treated using hot air guns. The surface of the glass exhibited a temperature range from 160 °C to 170 °C. The glass layer is quite thick, 4 mm, and all of the heat does not penetrate the glass to the polymer surface. A glass temperature of 170 °C ensures that EVA is above the melting point, *T*_m_ = 66.9 °C, and below the decomposition temperature, *T*_d_ = 215–385 °C, in the air environment [28,29]. Under these conditions, the EVA is soft, and the glass pulls easily from the surface of the PV module. The process of separating the glass required the bending of the PV module and the use of mechanical force, achieved through the utilization of a metal blade to pry apart the glass fragments. Throughout the process, no instances of EVA yellowing or residual gas release were recorded, indicating the absence of polymer degradation. The PV module with separated glass is shown in Figure 1b.

In the process of heating the PV module, it is feasible to remove the outermost layer of the backsheet as well, which is PET. It is possible after heating the backsheet to a temperature above 130 °C and mechanically pulling off the PET layer. Moreover, the separation of the PET layer from the backsheet also occurs when it is exposed to a solvent, as detailed in the next paragraph.

### 3.2. Chemical Treatment of EVA

A variety of organic solvents were used to swell the EVA copolymer and fractionate the photovoltaic module. The EVA copolymer exhibits the highest degree of swelling and partial dissolution in toluene, cyclohexane, xylene, tetrahydrofuran, o-dichlorobenzene, and trichloroethylene. In this study, toluene was selected due to its popularity, affordability, lower volatility, and reduced harm compared to chlorohydrocarbons. Nevertheless, a combination of xylene isomers exhibits comparable efficacy.

First, the effect of temperature on the rate of the delamination process was investigated. Each time, a sample of a 5 cm × 5 cm PV module and the same amount of 100 mL of solvent were used for the measurement. The system was stirred with a mechanical stirrer at a rotational speed of 500 rpm. Measurements were performed at three distinct temperatures, denoted as *T*_1_ = 25 °C, *T*_2_ = 35 °C, and *T*_3_ = 45 °C. In Figure 3, the dependence of the degree of PV module delamination on the duration of the EVA swelling process is presented.

It can be seen from the above data that the effect of temperature on the delamination process of PV modules is significant. For *T*_1_ = 25 °C, the delamination and swelling process of EVA ended after 50 min; for *T*_2_ = 35 °C, after 35 min; and for *T*_3_ = 45 °C, after 30 min. As the temperature increases, the delamination process becomes shorter. At higher temperatures, the process of solvent penetration into the polymer matrix is facilitated. The degree of polymer segment mobility in the solvent phase experiences a significant rise. EVA has a higher degree of swelling efficiency.

In the next step, an ultrasonic bath was used to test the effect of ultrasonics on the swelling process of the EVA copolymer and the delamination of PV modules. A sample of a 5 cm × 5 cm PV module and the same amount of solvent as in previous studies, 100 mL of toluene, were placed in the beaker. The process was carried out in two ways: without mechanical stirring and with 500 rpm mechanical stirring. The process was carried out until constant values of density and viscosity of the solvent solution and delamination of the photovoltaic module were established. Figure 4 shows the results compared to the process carried out at a constant temperature, *T* = 35 °C, with mechanical stirring of 500 rpm.

The EVA swelling and delamination of PV modules using the ultrasonic bath without mechanical stirring exhibited a less dynamic and slower behavior compared to the procedure performed at a constant temperature, *T* = 35 °C, with mixing. Behavior is attributed to two factors: first, the lack of stirring makes it more difficult to access the EVA and for the solvent to penetrate it, detaching fragments of the swelling polymer from the surface of the module. Moreover, the delamination process began at a slightly lower temperature of 24.3 °C. As depicted in Figure 4, the temperature exhibited an upward trend throughout the procedure. As the temperature increases, the delamination and swelling process becomes more active. The temperature of the solution in an ultrasonic bath increases in a nearly linear manner.

The most dynamic effect of EVA swelling and PV module delamination was achieved using both mechanical stirring and ultrasound, as shown in Figure 4. Ultrasound is thought to have two effects: firstly, it raises the temperature of the system, making it easier for the EVA to expand. Secondly, it helps the solvent penetrate the polymer chains and further breaks down the swollen EVA copolymer into smaller pieces. A delamination time of 35 min was achieved. The current time is similar to the delamination time achieved in a procedure conducted at a consistent temperature of *T* = 35 °C with agitation at 500 rpm.

The chemical delamination process can be divided into the following distinct steps, which are presented in Figure 5. During the initial stage (stage 1), there is an indication of swelling and fragmentation of the top laminating layer of silicon cells, as shown in Figure 5a. Stage 2 of the procedure entails removing the laminate layer, which causes the metal dust and crushed silicon cells to separate and fall into the solvent solution, as shown in Figure 5b. Suspension of metallic dust particles is obtained. Stage 3 involves the swelling and defragmentation of the inner laminating EVA layer that is bonded to the PVDF polymer. In stage 4, the backsheet undergoes a process of separation into two distinct components, namely, PVDF and PET, presented in Figure 5c,d, respectively. The backsheet separation is facilitated by the presence of a solvent and the application of mechanical forces generated through mixing.

During the chemical delamination process, the following material fractions are obtained: polymers, silicon cells, and metals. From the polymer fraction, the backsheet, separated into PET and PVDF, can be easily isolated from crushed EVA and PV cells with a diameter < 3 mm using a sieve with a mesh diameter of 6–10 mm. Both PET and PVDF exhibit resistance to swelling and dissolution upon exposure to toluene; however, they are separated from each other. PET is obtained in pure form after evaporation of the residual solvent, as will be confirmed by the analyses described in the next section. The purity of separated PVDF is comparatively lower than that of PET. The PVDF surface exhibits a coating of copolymer EVA residues. It occurs as a result of the interpenetration of PVDF and EVA polymer chains during the lamination process, which is conducted at an elevated temperature. The solvent used is unable to dissolve the EVA and separate it from the PVDF surface.

In the next step, the EVA and the crushed PV cells are separated, as illustrated in Figure 5a. Following the evaporation of the residual solvent, the separation of the PV cells from the EVA material occurs due to the disparity in density between these substances. For this purpose, the mixture of EVA and PV cells was placed in a glass beaker with water, as shown in Figure 6b. The density of EVA is lower than that of water and the polymer floats on the surface. A qualitative analysis was performed on the EVA that had been isolated and dried. Some of the recovered and dried PV cells contain a crosslinked EVA laminating polymer on their surface which has not been swollen and separated by etching with the solvent, toluene. The only way to remove the residue of laminate contamination is by pyrolysis or burning [28,29]. Figure 6c shows the PV cells after EVA burns in the flame of a gas burner.

### 3.3. Characteristics of Separated Materials

The recovered polymers were subjected to a range of characterization procedures to assess their structural composition, elemental composition, and thermal behavior.

#### 3.3.1. FTIR Spectroscopy

The chemical compositions of the recovered polymer samples were analyzed using Fourier transform infrared (FTIR) spectroscopy in attenuated total reflectance (ATR) mode. The FTIR results show that the encapsulating material is EVA (Figure 7) and that the backsheet layer is composed of PET (Figure 8) and PVDF (Figure 9). To determine the functional groups present in the recovered EVA, PET, and PVDF polymers, the material was analyzed in the spectral range from 4000 to 400 cm^−1^ and compared to the reference spectra of EVA, PET, and PVDF available in the literature [22,30]. The major vibrations of the EVA and literature sample are described and shown in Figure 6. Two bands at 2888 cm^−1^ and 2848 cm^−1^ are assigned to the symmetric and antisymmetric stretching bond of C–H in the −CH_2_ group. The band at 1735 cm^−1^ comes from the stretching bond of the carbonyl group, C=O. Two bands are present at 1468 cm^−1^ and 1370 cm^−1^, which is due to the bending of C−H. The EVA sample has a very concentrated band at 1233 cm^−1^, which indicates the presence of a C−O−C bond in the ester functional group. Lastly, there are two minor bands observed at 1018 cm^−1^ and 719 cm^−1^, which are attributed to the stretching of the C−O bond and the rocking vibration of the −CH_2_ group, respectively.

Furthermore, the broad peak between 3200 cm^−1^ and 3600 cm^−1^ corresponds to the vibration of the hydroxyl group −OH. The presence of moisture in the EVA is due to the production of acetic acid during deacetylation [31], as well as moisture permeating the laminating layer during use and the separation process. The FTIR spectrum of the investigated recovered material fully corresponds to the spectrum from the literature data for both the recycled and virgin polymers [30].

In the PET spectra, small bands were observed in the region of 2918 cm^−1^; these are symmetric and antisymmetric stretching bonds of C−H in the −CH_2_ group that were also present in the EVA. The larger band observed at 1712 cm^−1^ corresponds to the bond stretching of the carbonyl group, C=O, and at 1340 cm^−1^ to the bending vibrations of C-H bonds in a methylene group. The bands detected at 723 cm^−1^, 872 cm^−1^, and 1409 cm^−1^ were ascribed to modifications arising from the CH=CH aromatic ring. Additionally, 1094 cm^−1^ bands associated with symmetric C−O stretching vibrations and 1245 cm^−1^ bands associated with asymmetric C−O−C stretching vibrations in the ester group were identified. The FTIR spectrum of the PET recovered from the backsheet is in complete agreement with the spectrum found in the published data [22]. FTIR analysis confirms that the EVA from the laminating layer and PET recovered from the backsheet, during both the lamination process and the use of PV panels, as well as during the mechanical and chemical recycling processes, have not undergone degradation and have maintained their original structure.

The laminated PV module also includes a white plastic layer that was placed between the EVA and PET layers. According to the literature, this can be assumed to be a thin PVDF layer. Nevertheless, the FTIR spectrum obtained from the sample, presented in Figure 8, does not validate this hypothesis. On the recorded spectrum, characteristic bands for the EVA polymer can be observed at 1740 cm^−1^ due to the bond stretching of the carbonyl group, C=O, which is attributed to the ester group. Two bands at 1239 cm^−1^ indicate the asymmetric C−O−C stretching vibrations in the ester group, and at 1018 cm^−1^, they are associated with the stretching of the C−O bond. However, significantly more intense bands originating from the methylene −CH_2_ group can be detected in the spectrum. The symmetric and antisymmetric stretching bonds of C-H are associated with two bands located at 2914 cm^−1^ and 2847 cm^−1^, respectively. The presence of two bands at 1462 cm^−1^ and 1370 cm^−1^ can be attributed to the C−H bending. The rocking vibration of the −CH_2_ group occurs at 719 cm^−1^.

The FTIR spectrum indicates that the PVDF, which has been separated, contains leftover EVA polymer both on its surface and within its material. Neither mechanical nor chemical recycling methods achieved complete separation of the pure PVDF, compared to EVA and PET polymers. The PVDF spectrum does not fully match the spectrum seen in the literature [22] for the material extracted from the PV module. It has been verified that PVDF was obtained through recycling from the backsheet, but it is contaminated with EVA inclusions.

#### 3.3.2. Elemental Analysis

The molar percentages of carbon (C), hydrogen (H), nitrogen (N), and sulfur (S) in the separated EVA, PET, and PVDF polymers are presented in Table 1. The polymers lack nitrogen and sulfur atoms in their structure, and these elements were not identified in the investigation.

The C and H contents of the EVA and PET polymers showed a satisfactory level of agreement with the composition derived from their structure. This provides additional proof that the polymers, which were separated using mechanical and chemical methods, were successfully obtained in their pure form. However, the quantified carbon and hydrogen composition of the PVDF layer exceeds the calculated values. This suggests that the layer has been contaminated with another polymer, EVA, that contains a high concentration of C and H atoms.

#### 3.3.3. Differential Scanning Calorimetry

The results of the differential scanning calorimetry, DSC, measurements for EVA, PET and PVDF film samples recovered during the recycling of PV modules are shown in Figure 10. The measurements were performed over a wide range from −30 °C to 180 °C.

On the thermogram of the EVA, two overlapping endothermic peaks can be observed. The notable characteristic of these findings is the presence of two endothermic processes resulting from two groups of crystalline perfection. Published literature contains examples of this dual endothermic behavior [32,33,34]. The low-temperature endotherm corresponds to the melting of a group of imperfect, smaller crystallites, whereas the high-temperature endotherm is from the melting of larger, more regularly formed crystallites. The imperfect crystals are the result of the incorporation of branching and vinyl acetate comonomers into the polyethylene crystal lattice [33]. The peak temperature and total enthalpy of melting are summarized in Table 2.

In the temperature range studied, no thermal transitions are observed on the thermogram of the PET, whether related to glass transition, melting, or crystallization of the plastic. This is consistent with the literature because for PET, a melting peak between 170 and 270 °C, with a maximum of 256 °C, is observable [35].

The thermogram of the PVDF material exhibits two distinct temperature transitions. At a temperature, *T*_g_ = 32.3 °C, the glass transition is observed. The melting peak is between 60 °C and 115 °C, with a maximum at *T*_m_ = 101.4 °C and a melting enthalpy value of 78.1 J·g^−1^. The recorded temperatures of both phase transformations are lower than for the literature values, where the glass transition between 55 and 65 °C and the melting peak in the range from 110 to 180 °C were observed [35]. Moreover, the measured melting enthalpy value is higher than the literature value, which is 30 J/g for a degree of crystallinity of about 30% [35]. The results suggest that the recovered material is not pure PVDF, but rather consists of unseparated impurities in the form of EVA, as confirmed by FTIR spectrum analysis.

All three investigated polymers do not undergo thermal decomposition up to the tested temperature of 180 °C in an atmosphere of the inert gas, nitrogen.

## 4. Conclusions

The present research develops an alternative method for the separation of PV modules once they have reached the end of their lifespan. First, the PV module was heated to the temperature *T* = 170 °C, which allowed for the softening of the EVA and the mechanical separation of the glass layer. The removed glass exposed the EVA encapsulation layer and improved contact with the solvent in the chemical step. The most dynamic effect of EVA swelling and PV module delamination was achieved using mechanical stirring and ultrasound. A very short delamination time, *t* = 35 min, during the chemical process was achieved under mild temperature conditions, *T* = 35 °C.

FTIR spectroscopy, elemental analysis, and DSC thermal measurements confirmed that the recovered polymers were EVA applied as a laminating layer and PET derived from the backsheet. The laminated PV module also included a white plastic layer that was placed between the EVA and PET layers. According to the literature, this can be assumed to be a thin PVDF layer. However, the conducted study did not confirm the acquisition of a pure PVDF polymer through the proposed separation method as it is mainly contaminated with EVA.

Hence, it is concluded from various characterization techniques that the separated EVA and PET polymers show quite similar properties as those of commercial polymers. Therefore, they can be reused for encapsulation or as a backsheet and in other applications in the packaging and textile industries.

## Figures and Tables

**Figure 1 materials-17-00821-f001:**
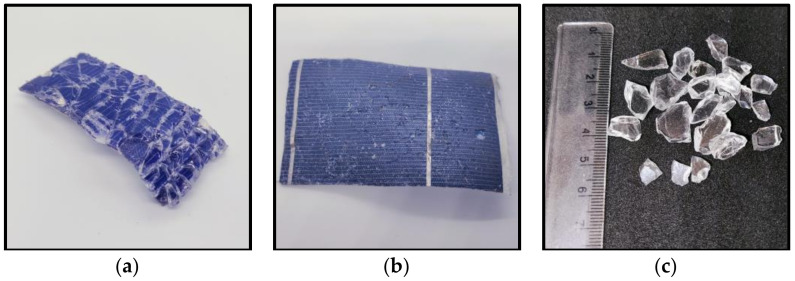
A photo of (**a**) an end-of-life PV module, (**b**) laminated PV samples after glass removal, and (**c**) removed glass.

**Figure 2 materials-17-00821-f002:**
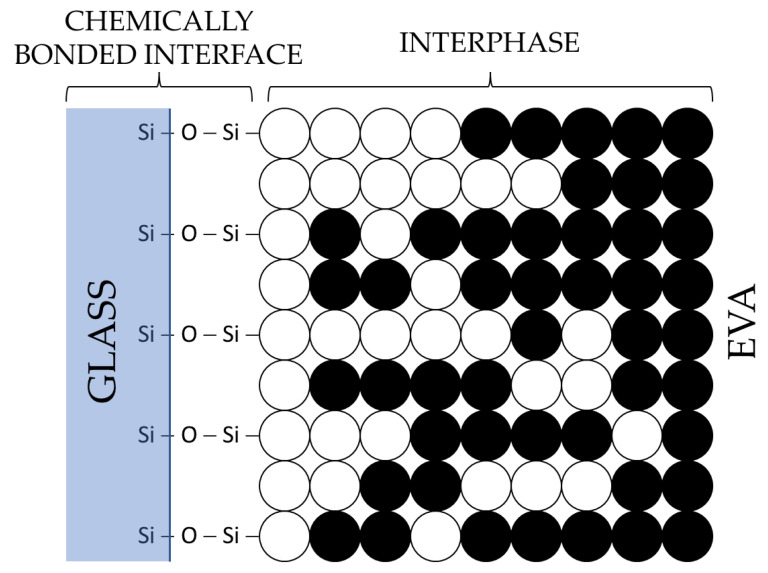
A schematic illustrating a potential interfacial configuration between EVA and glass, considering the inclusion of a silane adhesion promoter [27].

**Figure 3 materials-17-00821-f003:**
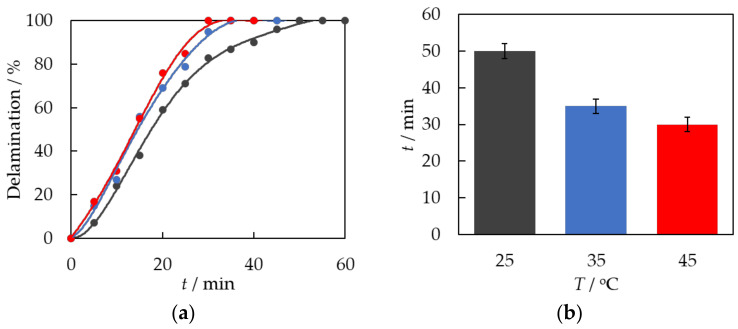
(**a**) The degree of PV module delamination as a function of time, *t*, during solvent exposure. dots—experimental data. A solid line is a guide for the eye. ●—*T*_1_ = 25 °C; ●—*T*_2_ = 35 °C; ●—*T*_3_ = 45 °C. (**b**) Delamination time of PV modules as a function of temperature.

**Figure 4 materials-17-00821-f004:**
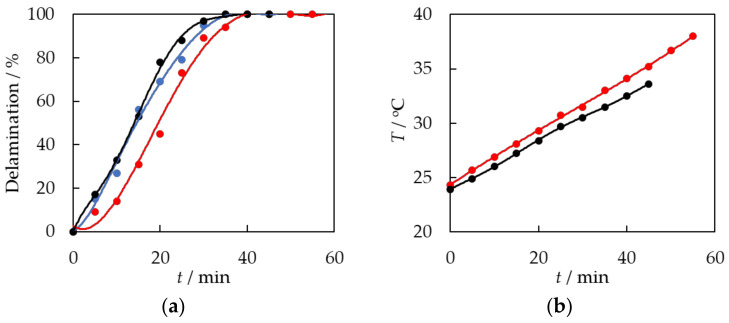
(**a**) The degree of PV module delamination as a function of time, *t,*, during solvent exposure. dots—experimental data. A solid line is a guide for the eye. ●—constant temperature, *T*_1_ = 25 °C; stirring at 500 rpm; ●—ultrasonic bath without stirring; ●—*T*_3_ = ultrasonic bath, stirring at 500 rpm. (**b**) Temperature changes in the ultrasonic bath during the delamination process: ●—ultrasonic bath without stirring; ●—*T*_3_ = ultrasonic bath, stirring at 500 rpm.

**Figure 5 materials-17-00821-f005:**
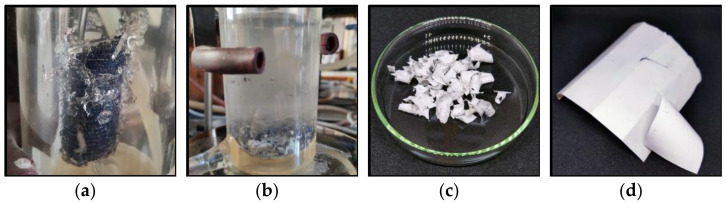
(**a**) Swollen and fragmented laminating EVA layer, (**b**) metallic dust suspension in the solvent, (**c**) PVDF, and (**d**) PET.

**Figure 6 materials-17-00821-f006:**
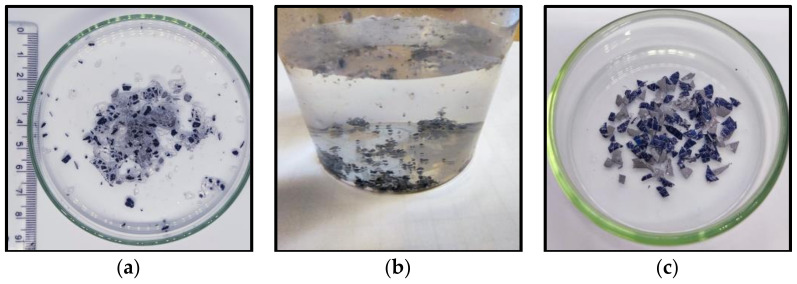
(**a**) A mixture of EVA and PV cells after chemical treatment, (**b**) the separation of EVA (on the surface) and PV cells (at the bottom of the beaker) in water, and (**c**) PV cells after burning.

**Figure 7 materials-17-00821-f007:**
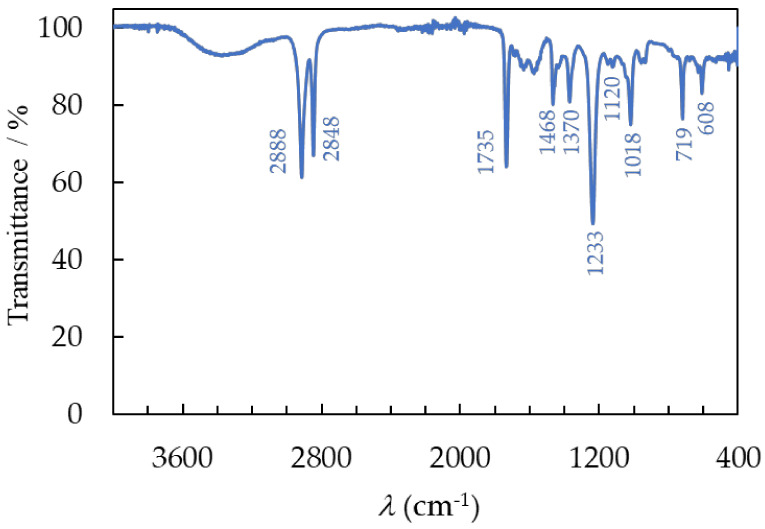
The FTIR spectra of recovered EVA after separation of PV module.

**Figure 8 materials-17-00821-f008:**
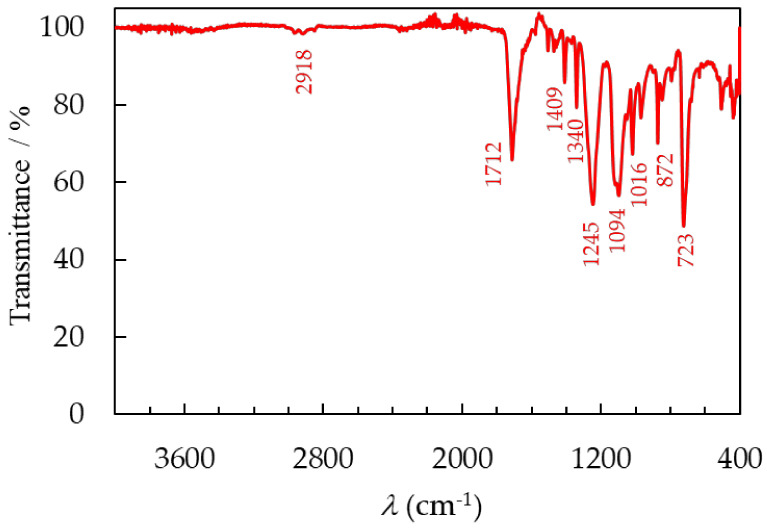
The FTIR spectra of recovered PET after separation of PV module.

**Figure 9 materials-17-00821-f009:**
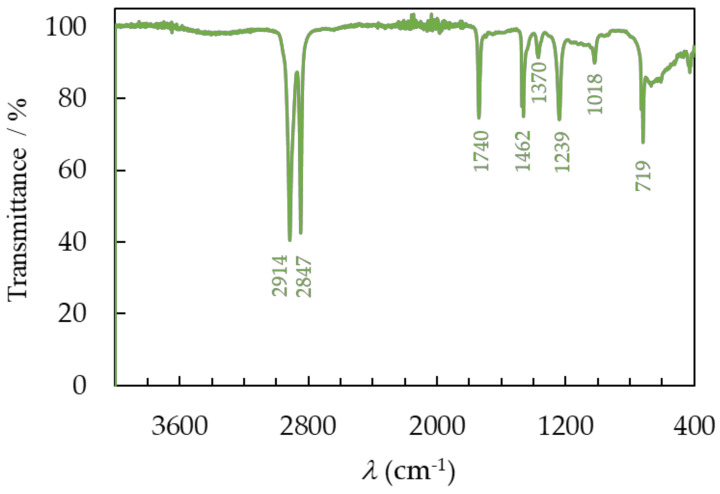
The FTIR spectra of recovered PVDF after separation of the PV module.

**Figure 10 materials-17-00821-f010:**
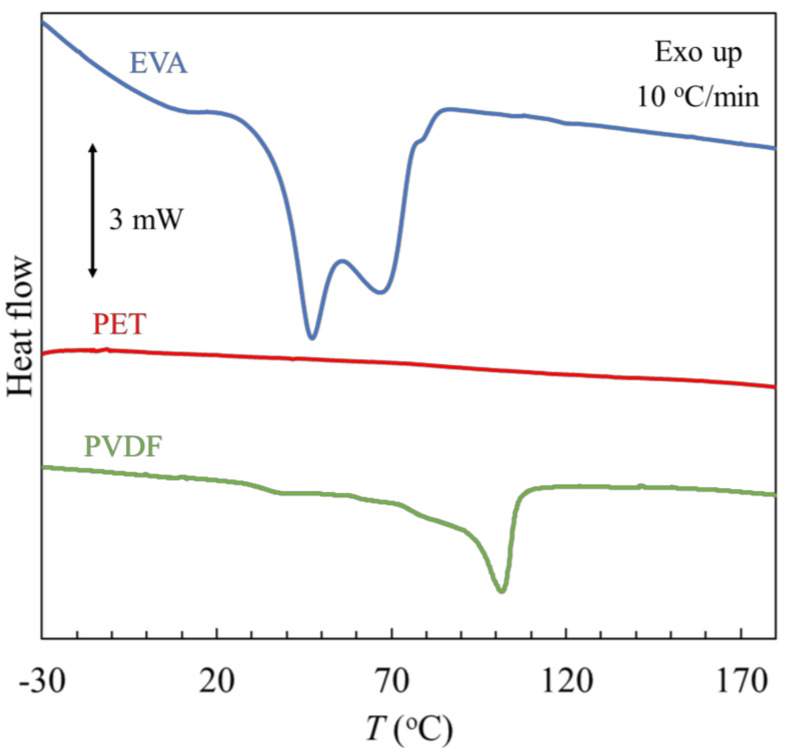
DSC analysis of separated polymers from the PV module.

**Table 1 materials-17-00821-t001:** Elemental analysis of C and H for EVA, PET, and PVDF.

Polymer	Element (%)	Observed	Calculated
EVA	C	71.63	70.58
	H	10.76	10.59
PET	C	63.10	62.50
	H	4.33	4.17
PVDF	C	47.23	37.51
	H	5.31	3.15

**Table 2 materials-17-00821-t002:** Thermal properties of separated polymers. Glass transition temperature, *T*_g_; heat capacity at glass transition, Δ_g_*C*_p_; melting temperature, *T*_m_; and enthalpy of melting, Δ_m_*H*.

Polymer	*T*_g_ (°C)	Δ_g_*C*_p_ (J·g^−1^·K^−1^)	*T*_m_ (°C)	Δ_m_*H* (J·g^−1^)
EVA	-	-	(1) 47.4, (2) 66.9 (1) 49, (2) 72 [34]	55.0
PVDF	32.3	0.57	101.4	78.1

Standard uncertainties *u* are as follows: *u*(*T*) *=* 0.3 °C; *u*(Δ_g_*C*_p_) = 0.05 J·g^−1^·K^−1^; *u*(Δ_m_*H*) = 0.5 J·g^−1^.

## Data Availability

The data presented in this study are available in the article and Appendix A.

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
