# Peer review of "Development of a Recycling Process and Characterization of EVA, PVDF, and PET Polymers from End-of-Life PV Modules"

_materials, 2024, doi:10.3390/ma17040821_

Round 1

Reviewer 1 Report

Comments and Suggestions for Authors

Comments

In this research, the authors developed a unique procedure for recycling PV modules, which provides an economical way of recycling and characterization of EVA, PVDF and PET polymers. In my view, the following issues need to be addressed before consideration for publication:

1. The article seems to lack explanations for Figures 5 and 6. Please provide additional descriptions.

2. There is an error of bookmark in the first paragraph of section 3.3.1, please correct it.

3. The authors compared the FTIR spectra of EVA and PET with references, but in the part of PVDF the authors did not do so. Please supplement, although it does not fully match the spectrum seen in the literature data.

4. The section of Elemental Analysis and Differential Scanning Calorimetry have been marked with the same number, please correct it.

5. Whether the thermogram of the PVDF is affected by unseparated impurities in the form of EVA?

Comments on the Quality of English Language

Minor editing of English language required

Author Response

Reviewer 1

In this research, the authors developed a unique procedure for recycling PV modules, which provides an economical way of recycling and characterization of EVA, PVDF and PET polymers. In my view, the following issues need to be addressed before consideration for publication:

  1. The article seems to lack explanations for Figures 5 and 6. Please provide additional descriptions.

Thank you very much for pointing out the lack of descriptions for the drawings. It has been appropriately corrected in the text.

  1. There is an error of bookmark in the first paragraph of section 3.3.1, please correct it.

It has been corrected.

  1. The authors compared the FTIR spectra of EVA and PET with references, but in the part of PVDF the authors did not do so. Please supplement, although it does not fully match the spectrum seen in the literature data.

Thank you very much for your attention. Indeed, the literature spectrum was not added in Figure 9. The spectrum is not fully consistent with the literature because, the pure PVDF polymer was not recovered, which is written in the manuscript.  Nevertheless, the literature reference is provided - references 22 and the characteristic bands can be compared. In addition, the literature spectra in the EVA and PET graphs were dropped because of the copyright of these figures. In our opinion, they are not necessary. In the paper, we presented our experimental spectra and provided a literature reference. Any reader of the paper can open the reference and compare the experimental spectra from the paper and the literature spectra. Thus, it will confirm that EVA and PET in pure form and PVDF contaminated with EVA were recovered.

  1. The section of Elemental Analysis and Differential Scanning Calorimetry have been marked with the same number, please correct it.

It has been corrected.

  1. Whether the thermogram of the PVDF is affected by unseparated impurities in the form of EVA?

Yes, exactly. EVA impurities have the effect of lowering the glass transition temperature of PVDF and lowering the melting point of PVDF compared to pure PVDF. However, only obtaining virgin EVA/PVDF composite materials of known composition would fully answer this question. How does the %wag EVA content affect the thermal properties of the EVA/PVDF material? Nevertheless, I did not find data on this subject in the literature. This could be a further research direction, especially for further applications of EVA/PVDF composite.

Reviewer 2 Report

Comments and Suggestions for Authors

First I would like to congratulate for the very good work on a very interesting topic.

Hereafter some minor observations:

1. pg. 3 line 113. Concerning softening temperature of EVA, it seems 170°C was measured on the glass. If possible, please indicate the softening temperature of EVA that is reported in literature which, from our experience, is much lower than 170°C.

2. pg. 8 line 326 a reference is missing

3. pg. 10 line 384 For elemental analysis I would have performed an analysis to determine Fluorine or Chlorine content: please check if it is possible to perform this analysis because it will support the evidence of alogenated layers in backsheet.

4. Finally please consider citing in the introduction one of these recent works on PV recycling:

Pietrogiovanni Cerchier, Katya Brunelli, Luca Pezzato, Claire Audoin, Jean Patrice Rakotoniaina, Teresa Sessa, Marco Tammaro, Gianpaolo Sabia, Agnese Attanasio, Chiara Forte, Alessio Nisi, Harald Suitner, Manuele Dabalà (2021). Innovative recycling of end of life silicon PV panels: ReSiELP. Detritus, 16, 41-47.

P. Cerchier, M. Dabalà, L. Pezzato, M. Tammaro, A. Zucaro, G. Fiorentino, G. Ansanelli, K. Brunelli Silicon-PV panels recycling: technologies and perspectives (2022). La Metallurgia Italiana, 16-26.

Author Response

Reviewer 2

First I would like to congratulate for the very good work on a very interesting topic.

Hereafter some minor observations:

  1. pg. 3 line 113. Concerning softening temperature of EVA, it seems 170°C was measured on the glass. If possible, please indicate the softening temperature of EVA that is reported in literature which, from our experience, is much lower than 170°C.

Thank you very much for the reviewer's comment. Yes, we agree with the reviewer. The melting point of EVA occurs in the range of 45 oC to 80 oC, depending on the VA content and the crosslinking degree. So the softening temperature is much lower than 170 oC. The temperature of 170 oC refers to the outer surface of the glass. The PV sample is heated to this temperature. However, the glass layer is quite thick (4mm), and all the heat does not penetrate the polymer surface. A glass temperature of 170 oC ensures that EVA is above its melting point, Tm = 66.9 oC, and below decomposition temperature, Td = 215-385 oC in the air environment. Under these conditions, the EVA is soft, and the glass pulls easily from the surface of the PV module. In addition, for virgin EVA at about 150 °C, there is an exothermic peak of crosslinking agent decomposition and a simultaneous crosslinking reaction. This is not observed in the thermogram shown in Fig. 10, because it is a recycled material, it is already crosslinked.

  1. pg. 8 line 326 a reference is missing

It has been corrected.

  1. pg. 10 line 384 For elemental analysis I would have performed an analysis to determine Fluorine or Chlorine content: please check if it is possible to perform this analysis because it will support the evidence of alogenated layers in backsheet.

I agree with the reviewer, additional fluoride analysis would be useful. However, at the moment it is not possible to determine fluorine using elemental analysis. The only elements that I can determine are C, H, N S and O. I cannot add a more extensive analysis in this manuscript.

  1. Finally please consider citing in the introduction one of these recent works on PV recycling:

Pietrogiovanni Cerchier, Katya Brunelli, Luca Pezzato, Claire Audoin, Jean Patrice Rakotoniaina, Teresa Sessa, Marco Tammaro, Gianpaolo Sabia, Agnese Attanasio, Chiara Forte, Alessio Nisi, Harald Suitner, Manuele Dabalà (2021). Innovative recycling of end of life silicon PV panels: ReSiELP. Detritus, 16, 41-47.

  1. Cerchier, M. Dabalà, L. Pezzato, M. Tammaro, A. Zucaro, G. Fiorentino, G. Ansanelli, K. Brunelli Silicon-PV panels recycling: technologies and perspectives (2022). La Metallurgia Italiana, 16-26.

Thank you for your suggestion, I will only cite one of the articles because they are similar in topic.